# Using P(Pressure)-T(Temperature) Path to Control the Foaming Cell Sizes in Microcellular Injection Molding Process

**DOI:** 10.3390/polym13111843

**Published:** 2021-06-02

**Authors:** Shia-Chung Chen, Che-Wei Chang, Chia-Yen Tseng, En-Nien Shen, Ching-Te Feng

**Affiliations:** 1R&D Center for Smart Manufacturing, Chung Yuan Christian University, Taoyuan 32023, Taiwan; landrick0130@gmail.com (C.-W.C.); chia.yen.tseng@gmail.com (C.-Y.T.); g10873063@cycu.edu.tw (C.-T.F.); 2R&D Center for Semiconductor Carrier, Chung Yuan Christian University, Taoyuan 32023, Taiwan ; Lynn.chu@gudeng.com

**Keywords:** microcellular injection molding, gas counter-pressure, gas holding/release time, dynamic molding temperature

## Abstract

Microcellular injection molding technology (MuCell) using supercritical fluid (SCF) as a foaming agent is one of the important green molding solutions for reducing the part weight, saving cycle time, and molding energy, and improving dimensional stability. In view of the environmental issues, the successful application of MuCell is becoming increasingly important. However, the molding process encounters difficulties including the sliver flow marks on the surface and unstable mechanical properties that are caused by the uneven foaming cell sizes within the part. In our previous studies, gas counter-pressure combined with dynamic molding temperature control was observed to be an effective and promising way of improving product quality. In this study, we extend this concept by incorporating additional parameters, such as gas pressure holding time and release time, and taking the mold cooling speed into account to form a P(pressure)-T(temperature) path in the SCF PT diagram. This study demonstrates the successful control of foaming cell size and uniformity in size distribution in microcellular injection molding of polystyrene (PS). A preliminary study in the molding of elastomer thermoplastic polyurethanes (TPU) using the P-T path also shows promising results.

## 1. Introduction

The development of microcellular foaming technology of polymers was initiated more than three decades ago [1,2]. In 1982, Martini et al. developed the foaming of polystyrene (PS) in solid-state using gas as a blowing agent in a batch process. About two decades later, Trexel Inc. successfully developed a microcellular injection molding process for commercial application and trademarked the process as MuCell [3] in 2001. The supercritical fluids (SCF) used in the MuCell process have characteristics of gas-like diffusivity, low viscosity, liquid-like density and have been considered as a physical foaming process. MuCell offers many advantages, such as lower melt viscosity, lower molding pressure (energy saving), and less weight. Foaming after melting filling plays a significant role in packing function leading to shrinkage reduction, warpage minimization, and elimination of sink marks, etc. The process characteristics and molded part properties have been studied earlier [4,5,6,7,8,9,10,11,12,13,14,15,16,17,18,19,20,21,22,23,24,25,26,27,28,29]. Despite its advantage, MuCell also faces technology challenges that hinder its wide application. One such challenge is silver-like swirl flow marks appearing on the part surface of MuCell. This is believed to be due to the combined effects of the partial foaming and fountain flow effects during the melt-filling process [8]. Uncontrollable and uneven foaming sizes are the other obstacle that the MuCell process encounters that further delay its usage in structural components such as automobile parts.

Improvement in the surface smoothness of the MuCell parts was proposed by earlier studies. These include the control of SCF content [10], material modification, use of in-mold decoration (IMD) process to decorate a film on the part surface, and co-injection molding leading to a solid skin and microcellular foamed core [11,12]. Dynamic mold temperature control [13,14,15,16,17,18,19] uses cavity surface coating or film-insert mold surface to delay the heat transfer on the cavity surface leading to an elevated mold temperature during the melt-filling stage. An alternate and effective way of solving the surface quality issue is the use of counter-pressure techniques, particularly the gas counter-pressure (GCP) technology [20,21,22,23,24], that have been reported to effectively improve the part surface roughness. Bledzki et al. [21] first reported a reduction in the part surface roughness (Rz) from 23 lm to 0.85 lm. Chen et al. [22] also reported that under suitable GCP control, the same gloss quality of the surface can be achieved compared to that of pure PS part molded by conventional injection molding. Chen et al. [24] further found that GCP not only improves the surface roughness but also provides an efficient way of controlling foaming cell size and influencing the relevant morphology. The pressure operation manner for controlling forming cell size and density was also proposed by Zou and Chen [25] using reciprocating compression and expansion by mold opening and closing. The improvement in cell density and the associated tensile strength are significant.

Systematic studies of gas counter-pressure effect on the MuCell process and conventional injection were carried out by Chen et al. [22,24,26,27,28]. The use of gas counter-pressure (GCP) at the melt front during the filling process can suppress the fountain flow effect and minimize the foaming bubbles being drawn to the part surface. This GCP effect was also useful in minimizing surface fiber orientation in injection and melt-powder separation in powder injection molding [27,28,29]. When the employed gas counter-pressure is higher than the critical pressure of the supercritical fluid, the dissolved SCF in melt remains in the SCF state and no foaming occurs in the filling stage. A schematic of GCP restricting foaming is presented in Figure 1.

When the gas counter-pressure is higher than the critical pressure, the melt may be retained as a single phase without any foaming as long as the counter-pressure applies. In addition, gas pressure holding for a certain period after melt filling can also restrict foaming and even result in no foaming if the long gas holding is applied.

In this study, a new concept of P(pressure)-T(temperature) (P-T) path control was employed for foaming control of MuCell parts (Figure 2). Most earlier studies on foaming control were conducted either by varying mold cavity temperatures [14,15,16,17,26] or by varying pressure alone [23,24] at a fixed mold temperature. Our study proposed an approach of sequential and/or simultaneous combined control of both pressure and temperature. We hope this approach can provide an effective and broad way of controlling foaming bubble sizes and the associated properties. Therefore, we extended this concept by incorporating additional parameters, namely, gas pressure holding time and release time, and considering the mold cooling speed (rather than mold temperature) to form a P-T path in the SCF PT diagram. Different parameter combinations constitute various PT paths. The foaming process is a pressure and temperature-driven reaction; therefore, different paths are expected to result in different foaming morphologies.

## 2. Experimental

An Arburg 420C injection molding machine equipped with a suitable screw for SCF dissolved in melt and a supercritical fluid generation/transportation system was used to conduct the MuCell process and inject molding foaming parts. A homemade gas pressure regulation unit installed with a high-frequency gas control valve was built and connected to the mold. The real-time gas pressure inside the mold cavity was monitored and adjusted by a Cavity pressure sensor (PRIAMUS 8102C). The system schematic is shown in Figure 3 [24]. For mold temperature control, two units were used sequentially, one for rapid heating and the other for rapid cooling, also shown in Figure 3. To achieve the cooling rate control, three mold blocks (one used single QC-10 aluminum, the other used single M333 steel, and the third block used both as composite) were designed and employed. This design enables the cooling rate control varying from 0.8 °C/s to10.9 °C/s in the mold temperature range of 90 °C to 120 °C. The molded part geometry and the associated mold cooling inert block are illustrated in Figure 4 [26]. Details have been described earlier [26]. The capability of gas counter-pressure control was also extended; instead of a fixed pressure value combined with the holding time, the gas pressure value could be decreased (released) to zero in a certain period (relief time) or via a multi-stage operation resulting in different pressure relief speeds. Various combinations of cooling speed and GCP relief speed created different P-T paths and possible variations of foaming morphology. The operation schematic is shown in Figure 2. The typically measured pressure and temperature profiles are illustrated in Figure 5 [24]. Before melt filling started, the pressure sensor measured the gas counter-pressure value-filled inside the cavity. Once melt advanced and passed the embedded pressure sensor, the pressure sensor measured the melt pressure which was higher than the GCP. It is because the melt needs an additional pressure drop to move ahead against the counter-pressure built at the melt front. Hence, the pressure curve shows a bump. Once the melt filled the cavity and stopped moving, the melt pressure returned to the GCP value. From the slope of the temperature variation, we could also calculate the cooling rate. The same situation was also applied to the gas pressure relief rate. The longer the relief time, the slower the GCP relief rate.

A general-purpose polystyrene resin (POLYREX PG-33, Chi-Mei Chemicals) was used in this study. Nitrogen was used as the SCF blowing agent and the gas source for counter-pressure build-up.

Relevant molding parameters for the MuCell, GCP, holding time, and release time are listed in Table 1. To measure the cell size and the solid frozen layer thickness, the field emission Scanning Electron microscope (Hitachi S-4800) was used. The average foamed size, as well as cell density, could then be calculated. A typical SEM image for measuring foaming bubbles is presented in Figure 6 [26]. The feasibility of the current approach was explored by a preliminary study for microcellular injection molding of thermoplastic elastomer TPU with/without GCP-based P-T control.

## 3. Results and Discussion

Let us first review the effects of GCP, mold temperature, and cooling rate on the foaming morphology. One of the results are presented in reference 24 reported earlier [24]. In the absence of GCP, the higher the mold temperature, the thinner the solid skin. However, it exhibits big foaming bubbles. When GCP was employed at low mold temperature, tiny and uniform foaming bubbles were formed, however, there existed a significantly thick, solid layer. This could decrease the weight reduction effect. When GCP was applied at higher mold temperatures, the morphology was of a thin solid layer and tiny and uniform bubbles. The foaming bubble seems to decrease in size with increasing GCP values. A recent study [26] showed the cooling effect (without GCP) on the foaming morphology. At a higher cooling speed and lower mold temperature, more tiny and uniform bubbles could be obtained. The results of GCP combined with the cooling effect at various GCP relief rates (or relief time) are presented in Figure 7. In this case, no holding time was employed when the filling process was complete. It does show that at a slower GCP relief rate, the foaming bobbles tend to be restricted and become smaller in size. The foaming restriction effect seemed more significant at higher mold temperatures when big foaming bubbles were frequently formed. The GCP holding was employed after melt-filling completion for a certain period (holding time) at various cooling rates, and the results are illustrated in Figure 8 and Figure 9, corresponding to mold temperatures 90 °C and 120 °C, respectively. The GCP holding extends the foaming restriction influence leading to improved foaming qualities. The foaming cell size and bubble density are quantitatively described in Figure 10 and Figure 11. Finally, the representative influence of foaming morphology under various parameter combinations (i.e., different P-T paths) is illustrated in Figure 12. The proposal using combinations of pressure-temperature variations in different ways is successfully evidenced. This offers a broader view of the foaming control philosophy for microcellular injection molding. To show the feasibility of further applications, a preliminary investigation was conducted on the MuCell process of thermoplastics elastomer, TPU. As shown in Figure 13, GCP-based P-T path control results in tiny and uniform foaming bubbles are indicated in Figure 13b. Nevertheless, the foaming is not easy to control in the regular injection molding process, as shown in Figure 13a.

## 4. Conclusions

In this study, a new method of using the P-T path to control foaming cell morphology in microcellular injection molding was developed. An approach of sequential and/or simultaneous control by a combination of both pressure and temperature was conducted. The operation parameter includes not only gas counter-pressure values but also gas holding time and gas relief rate. For temperature control, not only the mold temperature was varied but also the cooling rate control was employed by using different materials for cooling inserts. This approach shows promising results in foaming-cell size control. Under the current processing window, the average cell sizes range from 38 μm to 96 μm with deviations of about 10~15%. Similar results were also found for the MuCell process of thermoplastics elastomer TPU.

## Figures and Tables

**Figure 1 polymers-13-01843-f001:**
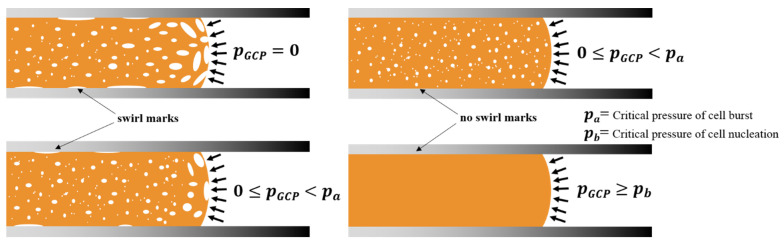
A schematic of gas counter-pressure (GCP) built at the melt front and its effect on the foaming process.

**Figure 2 polymers-13-01843-f002:**
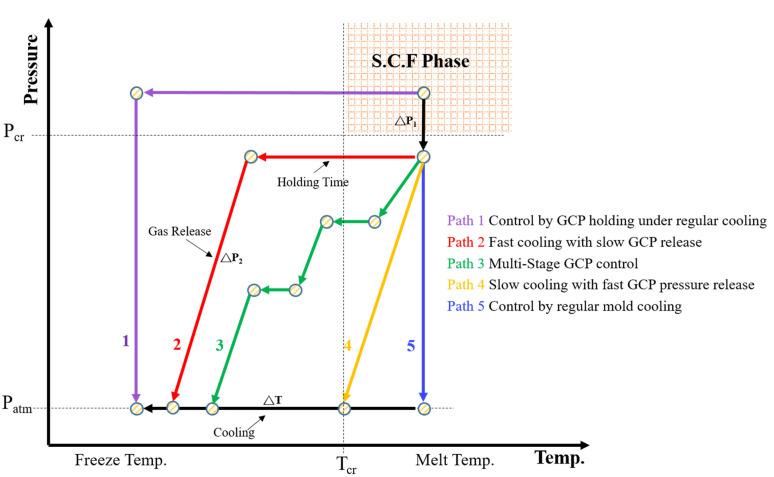
A schematic of various combinations of GCP parameters (pressure value, holding time, and relief time) and cooling speed leading to different P-T paths.

**Figure 3 polymers-13-01843-f003:**
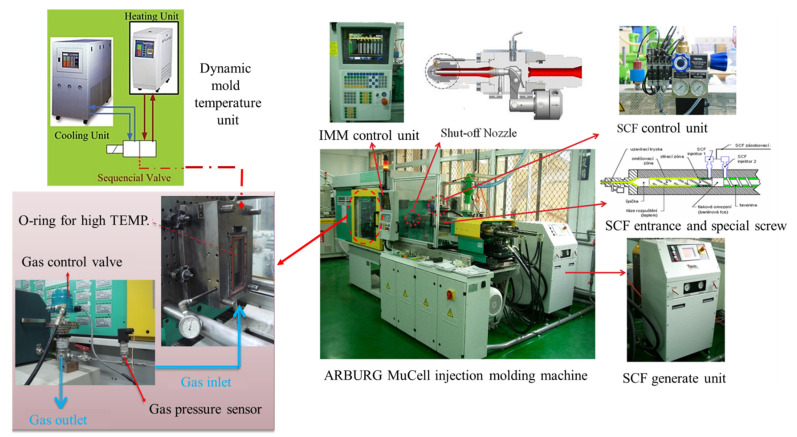
A schematic of MuCell injection molding Unit with a gas counter-pressure (GCP) regulation system and a dynamic mold temperature control system [24].

**Figure 4 polymers-13-01843-f004:**
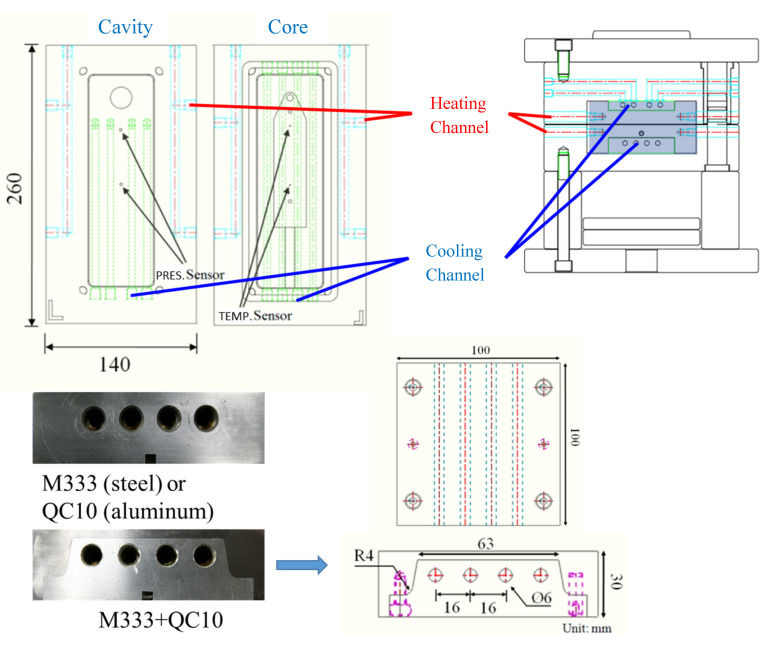
A mold schematic with three mold cooling blocks as well as part geometry [26].

**Figure 5 polymers-13-01843-f005:**
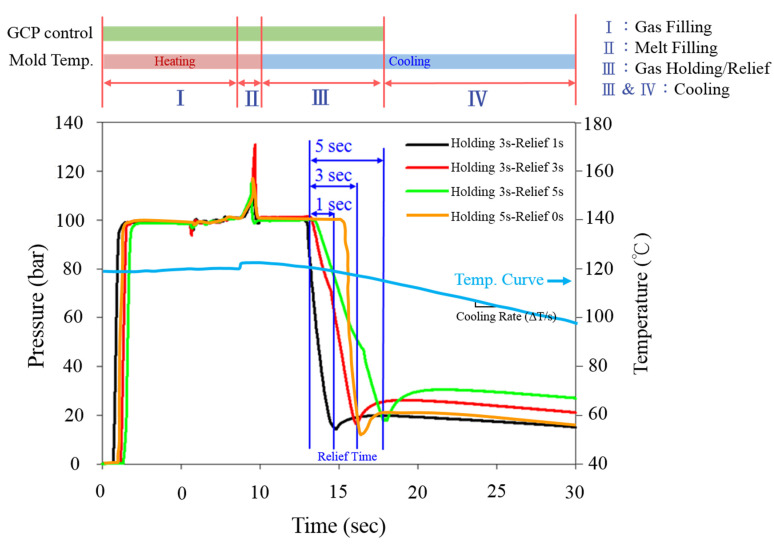
The typical measured pressure and temperature profiles.

**Figure 6 polymers-13-01843-f006:**
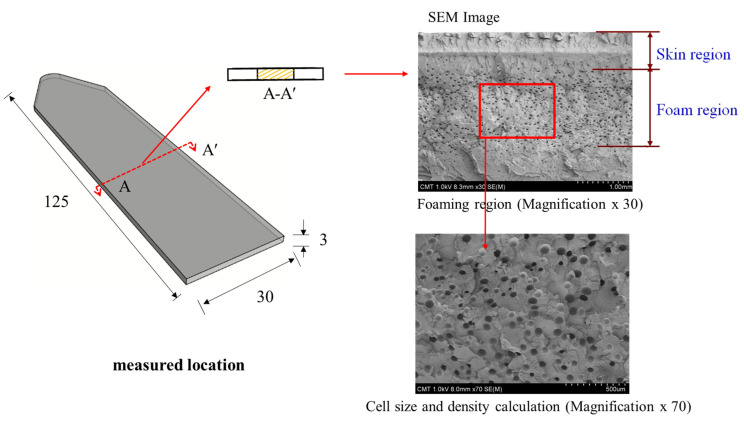
Measurement of the specimen at specified locations and the foaming bubble calculation of FE-SEM [26].

**Figure 7 polymers-13-01843-f007:**
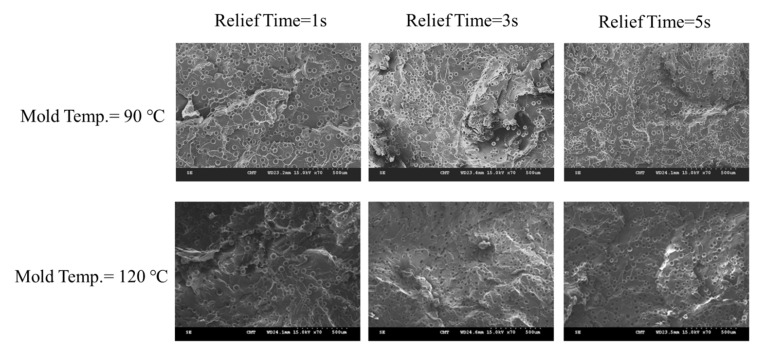
Effect of GCP release speed (relief time) on foaming morphology under two mold temperatures (90 °C and 120 °C).

**Figure 8 polymers-13-01843-f008:**
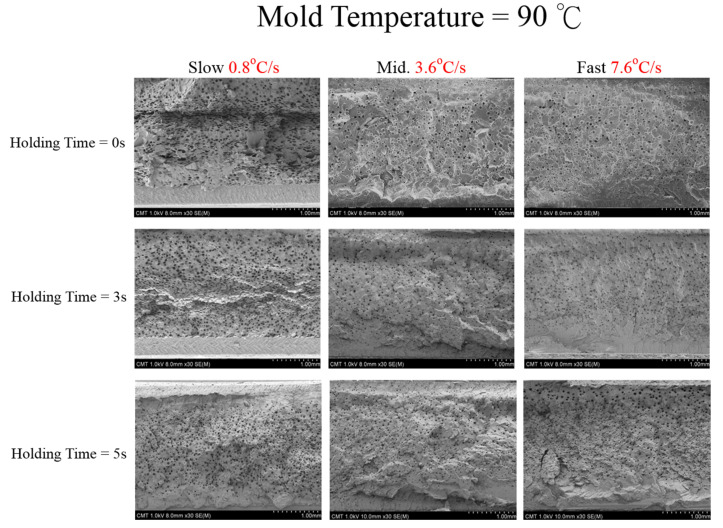
Effect of GCP holding time on foaming morphology under different cooling speeds at 90 °C mold temperature.

**Figure 9 polymers-13-01843-f009:**
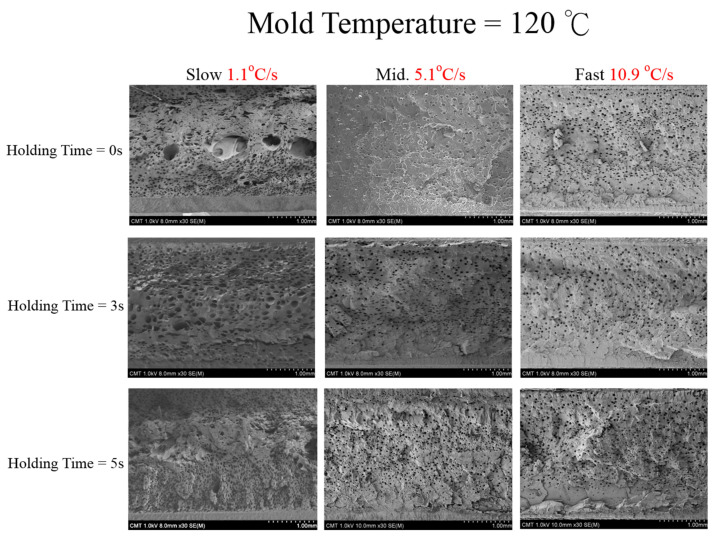
Effect of GCP holding time on foaming morphology under different cooling speeds at 120 °C mold temperature.

**Figure 10 polymers-13-01843-f010:**
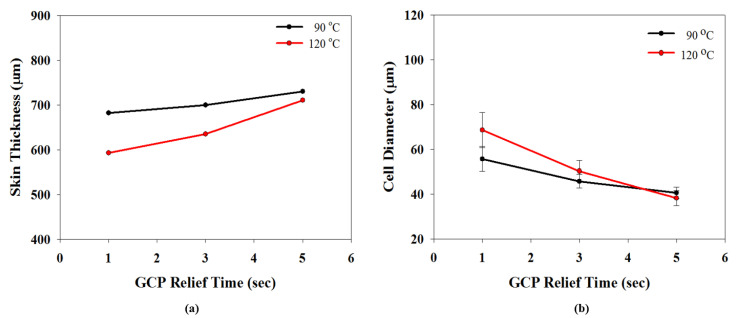
Effects of GCP relief time on sloid skin thickness and average foamed cell diameter. (**a**) Effects of GCP relief time on skin thickness, (**b**) effects of GCP relief time on foamed cell diameter.

**Figure 11 polymers-13-01843-f011:**
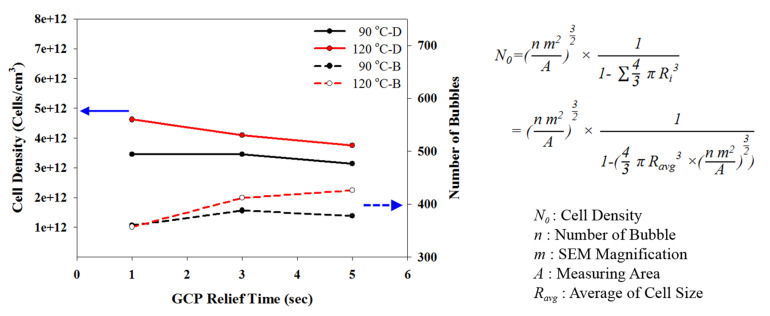
Effects of GCP relief time on foaming cell density. A schematic of cell density calculation from bubble numbers shown in the Scanning Electron Microscope photograph is also presented.

**Figure 12 polymers-13-01843-f012:**
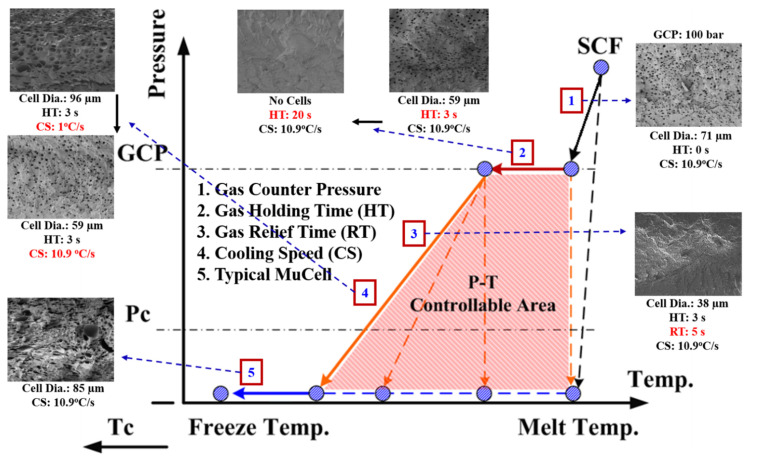
The representative influence of foaming morphology under various parameter combinations (i.e., different P-T paths) is illustrated.

**Figure 13 polymers-13-01843-f013:**
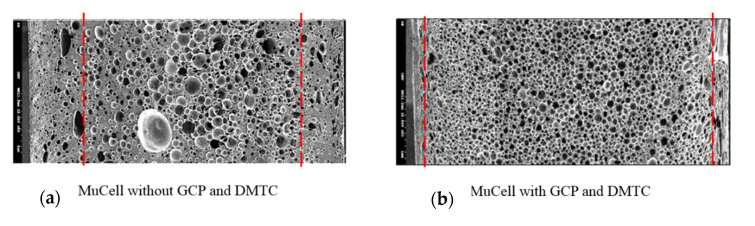
MuCell process of elastomer, thermoplastic polyurethanes (TPU). (**a**) Regular injection molding process without GCP control, (**b**) GCP-based PT path control results in tiny and uniform foaming bubbles.

**Table 1 polymers-13-01843-t001:** Molding conditions for microcellular molded part quality experiment using P-T path.

**Molding Conditions for Microcellular Injection Molding**	
Plastic resin	GPPS
Injection rate (cm^3^/s)	5
Specimen thickness (mm)	3
Melt temperature (°C)	210
Back pressure (Pa)	145
SCF ratio (%)	0.5
SCF injection time (s)	1.1
SCF injection quantity (Kg/h)	0.31
SCF injection pressure (Pa)	170
Delay time of valve gate (s)	5
**Mold Temperature Control**	
Mold temperature (°C)	120, 90
Cooling medium temperature (°C)	10
Cooling speed	Fast, medium, slow
**GCP Regulation Parameters**	
GCP (bar)	100
Holding Time (s)	0, 3, 5
Relief Time (s) (Holding Time 3 s)	1, 3, 5

## Data Availability

The data presented in this study are available on request from the corresponding author.

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
