# Peer review of "Using P(Pressure)-T(Temperature) Path to Control the Foaming Cell Sizes in Microcellular Injection Molding Process"

_polymers, 2021, doi:10.3390/polym13111843_

Round 1
Reviewer 1 Report
Dear,
Manuscript accept in present form.
Author Response
Dear reviewer,
Thank you for your kindly reply.
Reviewer 2 Report
The article entitled “Using P(pressure)-T(Temperature) path to Control the Foaming 2 Cell Sizes in Microcellular Injection Molding Process” contributed by Prof. Chen et al (Polymers-1235806) focuses on the influence of pressure-temperature path on the foaming effect. It is an interesting topic, and well suitable for the journal of POLYMERS, especially the special issue of “Precise Polymer Processing Technology”. The discussion of pressure-temperature path has great value for the researcher and the engineer in polymer processing field. Overall, the article was well prepared and well organized. This reviewer believes it can meet the standard of an academic paper, and recommends its publication. However, for the benefit of the reader, there are only two point which are better to be considered by the authors.
- In figure 12, the error bars have been marked in figure b, however, the title is “average”. Is it necessary? For figure a, it is not “average”? Please check it again.
- In introduction of background, a recent work, Polymers for Advanced Technologies, 2020, 31, 2136, is highly related to the controlling of cell size and morphology in the foaming injection molding. Maybe, it can help to expand the vision of the reader. The authors can determine to whether it should be mentioned by themselves.
In conclusion, it is a good contribution, and this reviewer suggests to accept it after some minor revisions.
Author Response
Dear reviewer,
Thank you for your suggestion.
For point 1.
In figure 12, the error bars have been marked in figure b, however, the title is “average”. Is it necessary? For figure a, it is not “average”? Please check it again.
We want to express that there will have a cell diameter variate even in single part here. The "average" in title and description was removed to prevent misleading the readers.
For point 2.
In introduction of background, a recent work, Polymers for Advanced Technologies, 2020, 31, 2136, is highly related to the controlling of cell size and morphology in the foaming injection molding. Maybe, it can help to expand the vision of the reader. The authors can determine to whether it should be mentioned by themselves.
That is an interesting work. We mentioned it in introduction and renew the cite list. Thank you for providing this information.